# ReMix: Reinforcement Routing for Mixtures of LoRAs in LLM Finetuning

## Abstract

Low-rank adapters (LoRAs) are a parameter-efficient finetuning technique that injects trainable low-rank matrices into pretrained models to adapt them to new tasks. Mixture-of-LoRAs models expand neural networks efficiently by routing each layer input to a small subset of specialized LoRAs of the layer. Existing Mixture-of-LoRAs routers assign a learned routing weight to each LoRA to enable end-to-end training of the router. Despite their empirical promise, we discover, both theoretically and empirically, that the routing weights are typically extremely imbalanced across LoRAs in practice, where only one or two LoRAs often dominate the routing weights. This essentially limits the number of effective LoRAs and thus severely hinders the expressive power of existing Mixture-of-LoRAs models. In this work, we attribute this weakness to the nature of learnable routing weights and rethink the fundamental design of the router. To address this critical issue, we propose a new router design that we call ***Re**inforcement Routing for **Mix**ture-of-LoRAs* (ReMix). Our key idea is using *non-learnable* routing weights to ensure all active LoRAs to be equally effective, with no single LoRA dominating the routing weights. However, such non-learnable routing weights make it infeasible to directly train routers via gradient descent. In response, we further propose an unbiased gradient estimator for the router and employ the reinforce leave-one-out (RLOO) technique to reduce the variance of the estimator. Our gradient estimator also enables to scale up training compute to boost the predictive performance of our ReMix. Extensive experiments demonstrate that our proposed ReMix significantly outperform state-of-the-art parameter-efficient finetuning methods under a small number of activated parameters.

## 1 Introduction

Parameter-efficient fine-tuning (PEFT) aims to reduce the number of trainable parameters while achieving strong task performance (e.g., He et al., 2022; Rücklé et al., 2020; Jie et al., 2023). Among PEFT methods, low-rank adapters (LoRAs, Hu et al., 2021) have become particularly prominent due to their simplicity and effectiveness. By injecting lightweight low-rank matrices into pretrained weight matrices, LoRAs allow downstream adaptation with a small fraction of trainable parameters, making them particularly attractive for resource-constrained settings and large-scale multi-task deployments.

Building on the success of LoRAs, researchers have proposed Mixture-of-LoRAs to further enhance parameter efficiency and expressive power (e.g., Huang et al., 2023; Wang et al., 2023; Tian et al., 2024; Zeng et al., 2025). The key idea is to route each input through a small pool of LoRAs per layer, thereby enabling specialization of LoRAs across different input distributions. Central to this framework is the router, which assigns routing weights across a pool of multiple LoRAs. Current approaches rely on learned routing weights, trained jointly with task objectives via gradient descent. In principle, such routers should flexibly allocate inputs across LoRAs and balance capacity usage.

Despite their empirical promise, we theoretically reveal a striking weakness of existing Mixture-of-LoRAs routers: routing weights can be extremely imbalanced, often with one or two LoRAs dominating the routing weights. Furthermore, we empirically observe that the imbalance even worsens during finetuning, where the effective number of LoRAs **drops to 1 quickly**. This essentially disables all other LoRAs, thereby limiting the expressive power of the mixture.

To address this critical limitation, we revisit the fundamental design of the router. Instead of relying on learned continuous weights that tend to result in extreme imbalance, we propose **Re**inforcement Routing for **Mix**ture-of-LoRAs (ReMix), which enforces a constant routing weights across all activated experts. This ensures that all active LoRAs contribute equally, avoiding collapse into a single dominant LoRA. Since non-learnable weights prevent direct training via backpropagation, we reformulate the router training problem as reinforcement learning (RL), where we view the supervised finetuning loss as the negative reward and the router as the policy model of RL. We then propose an unbiased, RLOO-based gradient estimator tailored for our proposed router. This unbiased estimator enables stable training and scales efficiently to large compute budgets, unlocking the full potential of mixture-based parameter-efficient finetuning. Our main contributions are as follows.

- **Theoretical insights on routing imbalance**: We theoretically reveal and empirically observe a fundamental limitation of routers: We observe that for each given input, often only one LoRA has a dominating routing weight that is close to one. This extreme imbalance essentially disables all other LoRAs and severely limits the expressive power of the model.

- **Simple yet effective router**: To address routing imbalance, we propose a new router design with a constant routing weight across all activated LoRAs. Our design does not introduce any additional inference cost over existing Mixture-of-LoRAs methods.

- **Reinforcement learning for router training**. To address the non-differentiability of our proposed router, we reformulate the router training problem as reinforcement learning and propose an unbiased, RLOO-based gradient estimator tailored for our proposed router.

- **Empirical evaluation**: Through extensive experiments across diverse benchmarks, we demonstrate that ReMix consistently outperforms state-of-the-art parameter-efficient finetuning methods under comparable parameter budgets.

## 2 EXTREME IMBALANCE OF ROUTING WEIGHTS

In this section, we analyze and reveal a critical limitation of existing Mixture-of-LoRAs routers: the extreme imbalance in routing weights assigned to different LoRAs. After introducing preliminaries in Section 2.1, we first make a fundamental theoretical analysis showing that the number of effective LoRAs per layer is severely limited. Then, we corroborate this finding with empirical evidence from our experiments.

### 2.1 PRELIMINARIES: MIXTURE OF LORAS

Mixture-of-LoRAs is a type of parameter-efficient adapter that enhances the capacity of large models using only a small number of LoRAs and a lightweight router to dynamically select the LoRAs for each input.

Let $D$ denote the hidden dimensionality of the model. Following prior work, we apply LoRAs to feedforward layers in the LLM, and all other layers are frozen. Let $\boldsymbol{x}^{(l)}, \boldsymbol{y}^{(l)} \in \mathbb{R}^D$ denote the input and the output of feedforward layer $l$ ($l = 1, \ldots, L$), respectively. Let $n$ denote the number of LoRAs we use in the mixture. Each LoRA $i = 1, \ldots, n$ is a linear map parameterized as a low-rank decomposition $\boldsymbol{B}_i^{(l)} \boldsymbol{A}_i^{(l)} \in \mathbb{R}^{D \times D}$, where $\boldsymbol{A}_i^{(l)} \in \mathbb{R}^{r \times D}$ and $\boldsymbol{B}_i^{(l)} \in \mathbb{R}^{D \times r}$ are learnable parameters, and $r \ll D$ is the rank of LoRAs. A *router* of a layer $l$ is a small neural network parameterized by a matrix $\boldsymbol{P}^{(l)} \in \mathbb{R}^{n \times D}$ that predicts a categorical distribution over $n$ LoRAs via the softmax operation:

$$\boldsymbol{\pi}^{(l)} := \mathrm{softmax}(\boldsymbol{P}^{(l)} \boldsymbol{x}^{(l)}) \in \mathbb{R}^n. \tag{1}$$

Here, $\pi_i^{(l)} := (\boldsymbol{\pi}^{(l)})_i$ represents the routing weight assigned to the $i$-th LoRA. Given the routing weights, the output of a typical Mixture-of-LoRAs layer is computed as:

$$\boldsymbol{y}^{(l)} := \boldsymbol{W}^{(l)} \boldsymbol{x}^{(l)} + \sum_{i=1}^{n} \pi_i^{(l)} \boldsymbol{B}_i^{(l)} \boldsymbol{A}_i^{(l)} \boldsymbol{x}^{(l)}. \tag{2}$$

where $\boldsymbol{W}^{(l)} \in \mathbb{R}^{D \times D}$ denotes the frozen weight of the layer $l$. This formulation intends to differentiably select a specialized subset of LoRAs for each given layer input $\boldsymbol{x}^{(l)}$.

## 2.2 THEORETICAL ANALYSIS

We make a fundamental theoretical analysis showing that the number of effective LoRAs is severely limited. Recall that the output of a Mixture-of-LoRAs layer is a weighted sum of the LoRA outputs, where the routing weights are typically normalized via a softmax function. While this design allows for end-to-end training, we show that it introduces a strong tendency for the router to concentrate most of the weight on only one or two LoRAs.

To quantify the effective number of LoRAs, we use the *effective support size* notion from information theory. For routing weights $\boldsymbol{\pi}^{(l)} \neq \mathbf{0}$, the effective support size (ESS) of $\boldsymbol{\pi}^{(l)}$ is defined as (Grendar, 2006)

$$\mathrm{ESS}(\boldsymbol{\pi}^{(l)}) := \frac{\left(\sum_{i=1}^n |\pi_i^{(l)}|\right)^2}{\sum_{i=1}^n |\pi_i^{(l)}|^2} = \left(\frac{\|\boldsymbol{\pi}^{(l)}\|_1}{\|\boldsymbol{\pi}^{(l)}\|_2}\right)^2. \tag{3}$$

The intuition of $\mathrm{ESS}(\boldsymbol{\pi}^{(l)})$ is that it measures the number of LoRAs with relatively large routing weights. For example, if $\boldsymbol{\pi}^{(l)}$ is one-hot, then we have $\mathrm{ESS}(\boldsymbol{\pi}^{(l)}) = 1$; if $\boldsymbol{\pi}^{(l)}$ is uniform over $n$ LoRAs, then we have $\mathrm{ESS}(\boldsymbol{\pi}^{(l)}) = n$. Note that $\mathrm{ESS}(\boldsymbol{\pi}^{(l)})$ concerns only about the utilization of LoRAs for each given input, not the overall utilization of each LoRA over the entire dataset. With the help of this notion of ESS, we formally state our theoretical observation in the following Theorem 1.

**Theorem 1** (imbalance of routing weights). *Suppose that the router parameter matrix $\boldsymbol{P}^{(l)}$ follows i.i.d. Gaussian initialization with variance $\sigma^2 > 0$ (e.g., Kaiming initialization, He et al., 2015). Then for any $0 < \delta < 1$, with probability at least $1 - \delta$, the effective support size of $\boldsymbol{\pi}^{(l)}$ is at most*

$$\mathrm{ESS}(\boldsymbol{\pi}^{(l)}) \leq \left(1 + \frac{1}{\exp\left(\frac{\delta\sigma\|\boldsymbol{x}^{(l)}\|_2}{\frac{3}{2}\sqrt{\frac{\pi}{\ln 3}}\ln n + \frac{1}{\sqrt{2\pi}2^{n-\log_2 n - 1}}} - \ln(n-1)\right)}\right)^2.$$

The proof of Theorem 1 is deferred to Appendix A.1. Our Theorem 1 shows that with high probability, only an extremely small number of LoRAs have relatively large routing weights. For instance, if $\sigma = 1$, and there are $n = 8$ LoRAs, and $\boldsymbol{x}^{(l)}$ is a Rademacher random vector in $\mathbb{R}^{D=1024}$, then our Theorem 1 shows that with probability at least 84.19%, **at most two** LoRAs have relatively large routing weights. Since each routing weight is a coefficient in front of each LoRA, a relatively small routing weight would essentially disable that LoRA. Moreover, those extremely small routing weights also vanish the gradient back-propagated to the corresponding LoRAs and consequently hinder the learning process of these LoRAs. Therefore, this phenomenon severely limits the expressive power and performance of the Mixture-of-LoRAs model.

## 2.3 EMPIRICAL ANALYSIS

To further validate our theoretical result in Theorem 1, we conduct a case study on the routing weights across LoRAs in MixLoRA (Li et al., 2024a), a popular Mixture-of-LoRAs method. Specifically, we track the routing weights of the last layer throughout the training process on the GSM8K dataset (a mathematical reasoning dataset) and compute the distributions and the ESS of the routing weights.

To visualize the distribution of routing weights, we plot a typical histogram of routing weights during finetuning, shown in Figure 1. We often observe that **only one LoRA** has a dominating routing weight close to one while **all other seven LoRAs** have negligibly small routing weights. The observation aligns with our Theorem 1

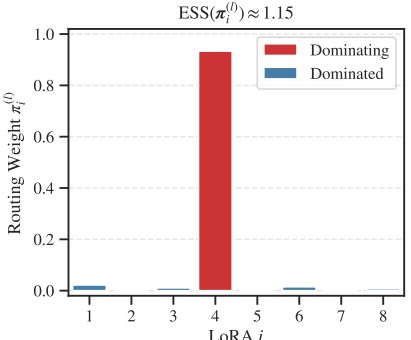

Figure 1: We often observe that **only one LoRA** has a dominating routing weight that is close to one.

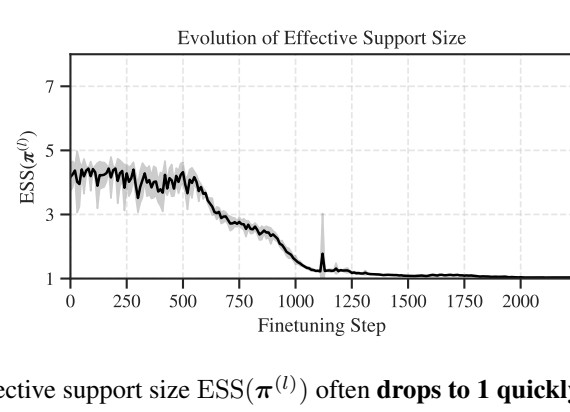

Figure 2: The effective support size $\mathrm{ESS}(\boldsymbol{\pi}^{(l)})$ often **drops to 1 quickly** during finetuning.

that the learned routing weights are indeed extremely imbalanced. The extremely limited number of effective LoRAs severely restricts the expressive power of the Mixture-of-LoRAs model.

To further study how the distribution of routing weights evolve over the finetuning process, we plot the ESS of the routing weights at each training step, as shown in Figure 2. In fact, the imbalance even worsens as the finetuning process progresses. We often observe that the effective support size $\mathrm{ESS}(\boldsymbol{\pi}^{(l)})$ often **drops to 1** quickly during finetuning. For instance, even though the ESS is around 4 at step 0, the ESS quickly decreases to 1 since only step 1000 and never increases thereafter.

These results highlight a fundamental limitation of current Mixture-of-LoRAs routers: despite the potential for increased expressivity via multiple LoRAs, the model essentially activates only an extremely small subset for each given input. This motivates our proposed method, which aims to ensure a more balanced and effective use of other available LoRAs.

## 3 PROPOSED METHOD: REMIX

In this section, we introduce our proposed method **Re**inforcement Routing for **Mix**ture-of-LoRAs (ReMix). First, we introduce the adapter architecture in Section 3.1. Then, we describe the finetuning procedure in Section 3.2 and the inference procedure in Section 3.3.

### 3.1 ADAPTER ARCHITECTURE

In this subsection, we introduce the adapter architecture of our proposed method ReMix.

Given layer input $\boldsymbol{x}^{(l)} \in \mathbb{R}^D$, we first produce an $n$-way categorical *routing distribution* $\boldsymbol{q}^{(l)} :=$ $\mathrm{softmax}(\boldsymbol{P}^{(l)}\boldsymbol{x}^{(l)}) \in \mathbb{R}^n_{\geq 0}$ over the $n$ LoRAs, where $\boldsymbol{P}^{(l)} \in \mathbb{R}^{n \times D}$ denotes the learnable parameter matrix of the router. Then, we use the routing distribution $\boldsymbol{q}^{(l)}$ to select the $k$ LoRAs $\mathcal{I}^{(l)} :=$ $\{i_1^{(l)}, \ldots, i_k^{(l)}\}$ to activate. The LoRA selection procedure differs between finetuning and inference, which we will describe later in Sections 3.2 & 3.3.

To address the extreme imbalance of routing weights in existing Mixture-of-LoRAs models (Section 2), we assign the a constant routing weight $\omega > 0$ to all the $k$ activated LoRAs and zero routing weights to all non-activated LoRAs. Formally, our routing weights $\boldsymbol{\pi}^{(l)}$ are defined as

$$\pi_i^{(l)} := \omega \mathbb{1}_{[i \in \mathcal{I}^{(l)}]} = \begin{cases} \omega, & \text{if } i \in \mathcal{I}^{(l)}, \\ 0, & \text{if } i \notin \mathcal{I}^{(l)}, \end{cases} \qquad i = 1, \ldots, n, \tag{4}$$

where $\omega > 0$ is a hyperparameter. Notably, our design ensures that $\mathrm{ESS}(\boldsymbol{\pi}^{(l)}) = k$, which is in stark contrast to existing learnable routing weights (Theorem 1). Finally, we compute the layer output $\boldsymbol{y}^{(l)} \in \mathbb{R}^D$ as a $\boldsymbol{\pi}^{(l)}$-weighted sum over $k$ activated LoRAs. Using the sparse nature of our routing

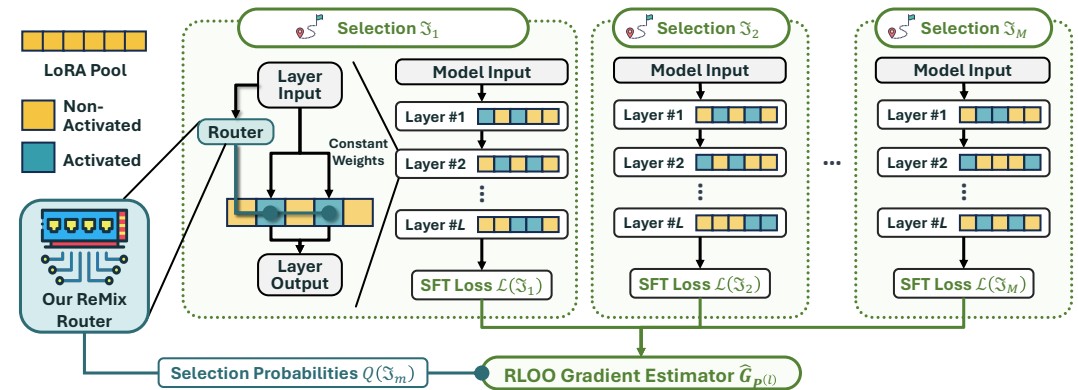

Figure 3: Finetuning procedure of our proposed ReMix.

weights $\boldsymbol{\pi}^{(l)}$, the computation of layer output $\boldsymbol{y}^{(l)}$ can be simplified as follows:

$$\boldsymbol{y}^{(l)} := \boldsymbol{W}^{(l)}\boldsymbol{x}^{(l)} + \sum_{i=1}^{n} \pi_i^{(l)} \boldsymbol{B}_i^{(l)} \boldsymbol{A}_i^{(l)} \boldsymbol{x}^{(l)} \tag{5}$$

$$= \boldsymbol{W}^{(l)}\boldsymbol{x}^{(l)} + \omega \sum_{j=1}^{k} \boldsymbol{B}_{i_j^{(l)}}^{(l)} \boldsymbol{A}_{i_j^{(l)}}^{(l)} \boldsymbol{x}^{(l)}. \tag{6}$$

### 3.2 FINETUNING PROCEDURE

In this subsection, we describe how to train our proposed ReMix during finetuning. Our finetuning workflow is illustrated in Figure 3.

Let $\mathfrak{I} := (\mathcal{I}^{(1)}, \ldots, \mathcal{I}^{(L)})$ denote the collection of activated LoRAs of the entire LLM for a given model input, and we call $\mathfrak{I}$ a *selection*. Let $\mathcal{L}(\mathfrak{I})$ denote the supervised finetuning (SFT) loss when activated LoRAs are $\mathfrak{I}$. Regarding LoRA parameters $\boldsymbol{A}_i^{(l)}$, $\boldsymbol{B}_i^{(l)}$, since the LLM output is differentiable w.r.t. LoRA parameters, we can simply use their gradients $\boldsymbol{G}_{\boldsymbol{A}_i^{(l)}} := \nabla_{\boldsymbol{A}_i^{(l)}} \mathcal{L}(\mathfrak{I})$, $\boldsymbol{G}_{\boldsymbol{B}_i^{(l)}} := \nabla_{\boldsymbol{B}_i^{(l)}} \mathcal{L}(\mathfrak{I})$ to train them.

Regarding router parameters, however, the LLM output is not differentiable w.r.t. router parameters $\boldsymbol{P}^{(l)}$ because routing weights $\pi_i^{(l)}$ are a constant hyperparameter $\omega$. Consequently, we cannot directly compute their gradients $\nabla_{\boldsymbol{P}_i^{(l)}} \mathcal{L}(\mathfrak{I})$ as it is not defined. To address this non-differentiability, we propose sampling each $\mathcal{I}^{(l)}$ from the corresponding routing distribution $\boldsymbol{q}^{(l)}$ so that $\mathbb{E}_{\mathcal{I}^{(l)} \sim \boldsymbol{q}^{(l)}}[\mathcal{L}(\mathfrak{I})]$ depends on router parameters $\boldsymbol{P}^{(l)}$. This enables $\mathbb{E}_{\mathcal{I}^{(l)} \sim \boldsymbol{q}^{(l)}}[\mathcal{L}(\mathfrak{I})]$ to be differentiable w.r.t. router parameters $\boldsymbol{P}^{(l)}$. Hence, we propose using $\boldsymbol{G}_{\boldsymbol{P}^{(l)}} := \nabla_{\boldsymbol{P}^{(l)}} \mathbb{E}_{\mathcal{I}^{(l)} \sim \boldsymbol{q}^{(l)}}[\mathcal{L}(\mathfrak{I})]$ as a *surrogate gradient* of $\boldsymbol{P}^{(l)}$. Formally, given the routing distribution $\boldsymbol{q}^{(l)} := \mathrm{softmax}(\boldsymbol{P}^{(l)}\boldsymbol{x}^{(l)})$, we sample $k$ LoRAs $(i_1^{(l)}, \ldots, i_k^{(l)}) \sim \boldsymbol{q}^{(l)}$ from $\boldsymbol{q}^{(l)}$ without replacement to compose the activated LoRA subset $\mathcal{I}^{(l)} := (i_1^{(l)}, \ldots, i_k^{(l)})$, where sampling without replacement ensures that the $k$ activated LoRAs are mutually distinct.

However, due to the exponentially many possibilities of $\mathfrak{I}$, it is computationally intractable to straightforwardly compute $\boldsymbol{G}_{\boldsymbol{P}^{(l)}} = \nabla_{\boldsymbol{P}^{(l)}} \mathbb{E}_{\mathcal{I}^{(l)} \sim \boldsymbol{q}^{(l)}}[\mathcal{L}(\mathfrak{I})]$ by definition. To address this intractability, we alternatively consider router training as a **reinforcement learning** (RL) problem, where we view the SFT loss $\mathcal{L}(\mathfrak{I})$ as the negative reward and the routers $\boldsymbol{q}^{(l)}$ as the policy model. With this alternative view, we are able to employ the policy gradient estimator in RL to estimate the surrogate gradient $\boldsymbol{G}_{\boldsymbol{P}^{(l)}}$. Formally, we independently sample $M$ selections $\mathfrak{I}_1, \ldots, \mathfrak{I}_M$, where $M$ represents the training compute budget. Write each selection as $\mathfrak{I}_m =: (\mathcal{I}_m^{(l)})_{l=1}^{L} =: ((i_{m,j}^{(l)})_{j=1}^{k})_{l=1}^{L}$ ($m = 1, \ldots, M$), where $\mathcal{I}_m^{(l)}$ denotes the ordered set of selected LoRAs at the $l$-th layer in the $m$-th selection $\mathfrak{I}_m$, and $i_{m,j}^{(l)}$ denotes the $j$-th selected LoRA at the $l$-th layer in the $m$-th selection $\mathfrak{I}_m$.

Due to sampling without replacement, the probability of each selection $\mathfrak{I}_m$ is

$$Q(\mathfrak{I}_m) := \prod_{l=1}^{L} \prod_{j=1}^{k} \frac{q_{i_{m,j}^{(l)}}}{1 - \sum_{j'=1}^{j-1} q_{i_{m,j'}^{(l)}}}. \tag{7}$$

Then, the REINFORCE policy gradient estimator (Willianms, 1988) for $\boldsymbol{G}_{\boldsymbol{P}^{(l)}}$ can be expressed as

$$\widetilde{\boldsymbol{G}}_{\boldsymbol{P}^{(l)}} := \frac{1}{M} \sum_{m=1}^{M} \mathcal{L}(\mathfrak{I}_m) \nabla_{\boldsymbol{P}^{(l)}} \log Q(\mathfrak{I}_m) \tag{8}$$

$$= \frac{1}{M} \sum_{m=1}^{M} \mathcal{L}(\mathfrak{I}_m) \sum_{j=1}^{k} \nabla_{\boldsymbol{P}^{(l)}} \log \frac{q_{i_{m,j}^{(l)}}}{1 - \sum_{j'=1}^{j-1} q_{i_{m,j'}^{(l)}}}. \tag{9}$$

Nevertheless, it is known that the vanilla REINFORCE estimator can have high variance (e.g., Kool et al., 2019). To further reduce the variance of the gradient estimator, we further employ the RLOO gradient estimator (Kool et al., 2019) to estimate the surrogate gradient $\boldsymbol{G}_{\boldsymbol{P}^{(l)}}$:

$$\widehat{\boldsymbol{G}}_{\boldsymbol{P}^{(l)}} := \frac{1}{M-1} \sum_{m=1}^{M} \left( \mathcal{L}(\mathfrak{I}_m) - \overline{\mathcal{L}} \right) \nabla_{\boldsymbol{P}^{(l)}} \log Q(\mathfrak{I}_m) \tag{10}$$

$$= \frac{1}{M-1} \sum_{m=1}^{M} \left( \mathcal{L}(\mathfrak{I}_m) - \overline{\mathcal{L}} \right) \sum_{j=1}^{k} \nabla_{\boldsymbol{P}^{(l)}} \log \frac{q_{i_{m,j}^{(l)}}}{1 - \sum_{j'=1}^{j-1} q_{i_{m,j'}^{(l)}}}, \tag{11}$$

where $\overline{\mathcal{L}}$ denotes the average SFT loss across the $M$ selections:

$$\overline{\mathcal{L}} := \frac{1}{M} \sum_{m=1}^{M} \mathcal{L}(\mathfrak{I}_m). \tag{12}$$

It can be shown that our RLOO gradient estimator is unbiased: $\mathbb{E}_{\mathfrak{I}_1, \ldots, \mathfrak{I}_m}[\widehat{\boldsymbol{G}}_{\boldsymbol{P}^{(l)}}] = \boldsymbol{G}_{\boldsymbol{P}^{(l)}}$.

## 3.3 INFERENCE PROCEDURE

In this subsection, we describe how our proposed ReMix selects the LoRAs to activate during inference. While it is possible to randomly sample the LoRAs like the finetuning procedure, here we propose a better, theoretically optimal approach to LoRA selection.

Our following Theorem 2 shows that the optimal strategy is in fact **top-$k$ selection** as long as the router is trained sufficiently well.

**Theorem 2** (optimality of top-$k$ selection). *Let $\mathcal{I}^{(l)*} = \{i_1^{(l)*}, \ldots, i_k^{(l)*}\}$ denote the optimal subset of LoRAs for a given model input. As long as the router $\boldsymbol{q}^{(l)}$ is trained sufficiently well such that*

$$\mathbb{P}_{\mathcal{I}^{(l)} \sim \boldsymbol{q}^{(l)}}[\mathcal{I}^{(l)} = \mathcal{I}^{(l)*}] > \frac{1}{2} \tag{13}$$

*then the LoRAs $i$ with top-$k$ $q_i^{(l)}$ are guaranteed to constitute the best subset $\mathcal{I}^{(l)*}$:*

$$\underset{i=1}{\overset{n}{\arg\mathrm{top}_k}} \, q_i^{(l)} = \mathcal{I}^{(l)*}. \tag{14}$$

The proof of Theorem 2 is deferred to Appendix A.2. Notably, our Theorem 2 shows that when sampling yields the optimal subset with probability above $50\%$, then top-$k$ selection substantially improves this probability to $100\%$. Intuitively speaking, as long as the router is trained sufficiently well, then the optimal choices of LoRAs are in fact those $i$ with top-$k$ $q_i^{(l)}$. Motivated by Theorem 2, we employ top-$k$ LoRA selection (instead of random sampling) during inference:

$$\mathcal{I}^{(l)} = \{i_1^{(l)}, \ldots, i_k^{(l)}\} := \underset{i=1}{\overset{n}{\arg\mathrm{top}_k}} \, q_i^{(l)}. \tag{15}$$

Table 1: Comparison with existing parameter-efficient finetuning methods. Our ReMix consistently outperforms all baseline methods while maintaining strong parameter efficiency.

| Type | Method | GSM8K | | HumanEval | | ARC-c | | Average | |
|---|---|---|---|---|---|---|---|---|---|
| | | Accuracy | Params | Pass@1 | Params | Accuracy | Params | Accuracy | Params |
| No Tuning | Zero-Shot | 04.78 | N/A | 13.41 | N/A | 22.03 | N/A | 13.41 | N/A |
| | Few-Shot | 55.95 | N/A | 17.68 | N/A | 81.36 | N/A | 51.66 | N/A |
| Prefix Injection | Prefix Tuning | 02.65 | 0.034B | 00.00 | 0.034B | 28.47 | 0.004B | 10.37 | 0.024B |
| | Prompt Tuning | 04.70 | 0.000B | 26.22 | 0.000B | 23.73 | 0.000B | 18.22 | 0.000B |
| | P-Tuning | 34.19 | 0.001B | 27.44 | 0.001B | 43.05 | 0.001B | 34.89 | 0.001B |
| Weight Modulation | $(IA)^3$ | 08.57 | 0.001B | 31.10 | 0.001B | 23.39 | 0.001B | 21.02 | 0.001B |
| | LoRA | 59.21 | 0.168B | 26.83 | 0.084B | 83.05 | 0.084B | 56.36 | 0.112B |
| | DoRA | 55.34 | 0.043B | 31.10 | 0.169B | 83.39 | 0.169B | 56.61 | 0.127B |
| | rsLoRA | 62.47 | 0.042B | 28.66 | 0.021B | 82.71 | 0.021B | 57.95 | 0.028B |
| Mixture | VB-LoRA | 34.27 | 0.677B | 29.27 | 0.673B | 23.73 | 0.674B | 29.09 | 0.675B |
| | MixLoRA | 61.87 | 0.068B | 28.05 | 0.116B | 82.37 | 0.119B | 57.43 | 0.101B |
| | HydraLoRA | 62.47 | 0.092B | 20.12 | 0.079B | 82.71 | 0.082B | 55.10 | 0.084B |
| | ReMix (Ours) | **65.66** | 0.106B | **32.93** | 0.090B | **83.73** | 0.016B | **60.77** | 0.070B |

## 4 EXPERIMENTS

### 4.1 EXPERIMENTAL SETUP

**Baselines.** We comprehensively compare our proposed ReMix against various types of baseline methods. (i) No tuning methods: testing the base LLM directly under zero-shot and few-shot prompting. (ii) Prefix injection methods: Prefix Tuning (Li & Liang, 2021), Prompt Tuning (Lester et al., 2021), and P-Tuning (Liu et al., 2021b). (iii) Weight modulation methods: $(IA)^3$ (Liu et al., 2022), LoRA (Hu et al., 2021), DoRA (Liu et al., 2024), and rsLoRA (Kalajdzievski, 2023). (iv) Mixture methods: VB-LoRA (Li et al., 2024b), MixLoRA, (Li et al., 2024a), and HydraLoRA (Tian et al., 2024). For each baseline method, we perform a hyperparameter search and report the best results.

**Datasets & evaluation metrics.** We finetune the base LLM and evaluate them on a diverse set of benchmarks, including GSM8K (Cobbe et al., 2021) to evaluate mathematical reasoning capabilities, HumanEval (Chen et al., 2021) to evaluate code generation capabilities, and ARC-c (Clark et al., 2018) to evaluate knowledge recall capabilities. For HumanEval, since HumanEval does not contain a training set, we follow Tian et al. (2024) to finetune the base LLM on CodeAlpaca (Chaudhary, 2023) and report the Pass@1 metric on HumanEval. For all other datasets, we finetune the base LLM on their training split and report the accuracy metric on their test split. In this work, we use Llama 3 8B (Dubey et al., 2024) as the base LLM. Besides that, we also report the number of activated parameters (in billion, B) under the best-performing hyperparameters.

**Implementation details.** We train all methods using the same number of epochs, learning rate schedule, gradient accumulation steps and machine type. All methods are trained using the LLaMA-Factory (Zheng et al., 2024) framework and evaluated using the OpenCompass (Contributors, 2023) framework. For the no-tuning few-shot method, we use 4 shots for GSM8K and HumanEval and 5 shots for ARC-c.

### 4.2 MAIN RESULTS

We evaluate the performance of various fine-tuning strategies on three representative tasks: HumanEval (code generation), GSM8K (math reasoning), and ARC-c (knowledge recall). As shown in Table 1, our ReMix consistently outperforms all baselines across these benchmarks while maintaining strong parameter efficiency.

From a performance standpoint, ReMix surpasses all baseline methods, achieving an average accuracy improvement of 2.82 over the strongest competing approach. Specifically, ReMix outperforms the best Prefix Injection baseline by a substantial 25.88, the best Weight Modulation baseline by 2.82, and the strongest Mixture competitor by 3.34 on average across the three tasks. On Hu-

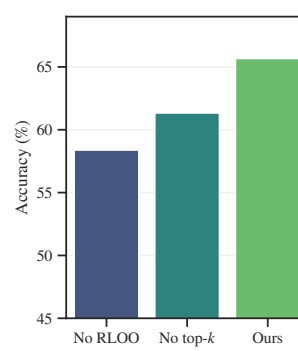

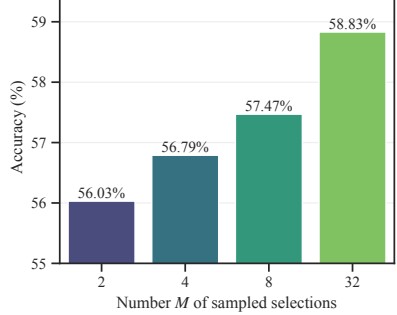

(a) Both of our proposed components RLOO and top-$k$ selection contribute significantly to the strong performance of our ReMix.

(b) Our proposed ReMix can further benefit from scaling up the training compute while existing methods are not able to.

Figure 4: Additional experimental analyses.

manEval, ReMix achieves a Pass@1 of 32.93, outperforming the best baseline, (IA)[3], by 1.83. For GSM8K, ReMix attains an accuracy of 65.66, showing a clear gain of 3.19 over the best competitors (rsLoRA and HydraLoRA). On ARC-c, ReMix reaches 83.73, exceeding the best-performing low-rank method DoRA by 0.34. These results highlight the consistent advantages of our reinforcement-trained router across diverse task types. Notably, within the Mixture methods, ReMix provides consistent improvements, suggesting that reinforcement-guided, balance-aware routing enhances both reasoning-intensive tasks (e.g., GSM8K) and generation tasks (e.g., HumanEval), while preserving strong retrieval performance on ARC-c.

In terms of parameter efficiency, ReMix achieves these performance gains with a competitive budget of only 0.070B trainable parameters. Compared to other mixture methods, this represents a 90% reduction relative to the most parameter-heavy baseline VB-LoRA (0.675B), and a 31% reduction compared to the most effective baseline MixLoRA (0.101B). Even when compared to the lightweight rsLoRA (0.028B), ReMix delivers a +2.82 average-accuracy improvement at the cost of only 0.042B more parameters, demonstrating a superior accuracy-to-parameter trade-off. Overall, these results confirm that reinforcement-guided mixture routing achieves state-of-the-art accuracy with minimal and often reduced parameter overhead.

### 4.3 ABLATION STUDIES

To understand the contributions of the key components in our proposed ReMix (i.e., RLOO for router training and top-$k$ LoRA selection for inference), we conduct ablation studies on GSM8K comparing its performance against the ablated variants with each component removed. The results are presented in Figure 4a, which visualizes the accuracy achieved by different configurations.

From Figure 4a, we observe that our full ReMix method achieves the highest accuracy among all ablated variants. When removing the RLOO from our finetuning procedure ReMix (No RLOO), we observe a significant drop in accuracy compared to the full ReMix, indicating that RLOO plays a crucial role in enhancing the model's performance. Similarly, disabling the top-$k$ LoRA selection (No top-$k$) also results in lower accuracy than the complete ReMix, demonstrating the importance of this component in optimizing the model performance. These findings underscore the value of integrating both RLOO and top-$k$ selection into our ReMix method.

### 4.4 TRAINING EFFICIENCY

In this subsection, we study the training efficiency of our proposed method. Note that MixLoRA can be regarded as an ablated variant where our reinforcement router is replaced with an ordinary learnable router. Hence, we compare our ReMix and MixLoRA under comparable training time to show the training efficiency of our proposed ReMix. The results are presented in Table 2.

As shown in Table 2, our ReMix achieves an accuracy of 56.03% with a training time of 9.87 seconds per step, while MixLoRA achieves an accuracy of 50.34% in 8.95 seconds per step. Although our ReMix consumes only 10% more training time than MixLoRA, it yields a substantial improvement of 5.69 percentage points in accuracy. This demonstrates that our ReMix still retains strong performance even under small training compute budget.

Table 2: Our ReMix significantly outperforms MixLoRA even under similar training time.

| Method | Time | Accuracy |
|---|---|---|
| MixLoRA | 8.95 s | 50.34 |
| ReMix (Ours) | 9.87 s | 56.03 |

### 4.5 BENEFITS FROM TRAINING COMPUTE SCALING

Since our ReMix incorporates RL-based gradient estimator, we can effectively scale up training compute by increasing the number $M$ of sampled selections. To evaluate how training compute scaling benefits our ReMix, we examine its performance under varying numbers $M$ of sampled selections. As shown in Figure 4b, increasing $M$ from 2 to 32 leads to a steady improvement in accuracy, rising from 56.03% to 58.83%. This indicates that our ReMix effectively leverages additional computational resources to enhance its performance. Notably, the consistent gains observed across different scales suggest that further increases in $M$ could yield even better results. This demonstrates that ReMix offers a favorable trade-off between training efficiency and performance. In stark contrast, existing methods do not benefit from similar scaling, underscoring the unique advantage offered by ReMix in utilizing increased training compute to achieve improved outcomes.

## 5 RELATED WORK

Parameter-efficient fine-tuning (PEFT) aims to reduce the number of trainable parameters while achieving strong task performance. Due to the page limit, please refer to Appendix B for related work on general PEFT. More recent efforts in PEFT have explored new multi-LoRA architectures that go beyond single low-rank adapters by explicitly restructuring how multiple LoRA modules are organized and combined, offering advantages on complex data distributions. LoraHub (Huang et al., 2023) introduces a dynamic composition framework that integrates multiple LoRAs at the architectural level, enabling cross-task generalization without retraining by assembling adapters into a unified pipeline. MultiLoRA (Wang et al., 2023) modifies the structural initialization of LoRA subspaces and horizontally expands adapters across layers, thereby mitigating the dominance of top singular vectors and achieving more balanced representations in multi-task learning. HydraLoRA (Tian et al., 2024) departs from the symmetric LoRA design and proposes an asymmetric architecture that decouples the projection and update pathways, substantially improving parameter and training efficiency. Beyond linear compositions, S'MoRE (Zeng et al., 2025) integrates LoRA with mixture-of-experts style routing by hierarchically decomposing expert weights into low-rank residual components and routing them through a structured multi-layer architecture. Meanwhile, LoRA-Flow (Wang et al., 2024) rethinks the architecture for generative tasks by embedding a lightweight, token-level fusion gate that dynamically modulates multiple LoRAs during inference, and MultLFG (Roy et al., 2025) introduces a frequency-aware fusion mechanism that structurally guides LoRA composition across denoising steps.

## 6 CONCLUSION

In this paper, we investigate the problem of imbalanced routing weights that hinder effective LoRA utilization, and propose a reinforcement-based router design named ReMix to address this problem. Our theoretical and empirical analysis shows that existing methods relying on learnable routing weights inherently lead to such imbalance, severely limiting the effective utilization of diverse LoRA knowledge. To overcome this limitation, we replace learnable routing weights with non-learnable balanced assignments, and introduce an unbiased gradient estimator with RLOO variance reduction to enable scalable and stable training under non-differentiable settings. Extensive experiments across diverse benchmarks demonstrate that our ReMix consistently outperforms state-of-the-art parameter-efficient finetuning methods, achieving superior predictive power and computational efficiency.

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

CONTENTS

## A THEORETICAL PROOFS

### A.1 PROOF OF THEOREM 1

Before stating our proof of Theorem 1, we present a few technical lemmata that we will employ.

Let $\varphi(z) := \frac{1}{\sqrt{2\pi}}\mathrm{e}^{-x^2/2}$, $\Phi(z) := \int_{-\infty}^{z} \varphi(x)\,\mathrm{d}x$, and $\overline{\Phi}(z) := 1 - \Phi(z)$ $(z \in \mathbb{R})$ denote the probability density function, the cumulative distribution function, and the complementary cumulative distribution function of the standard Gaussian distribution $\mathcal{N}(0,1)$, respectively.

**Lemma 3** (a Gaussian gap estimate). *For every $z \in \mathbb{R}$ and $\alpha > 0$,*

$$\Phi(z+\alpha) - \Phi(z) \leq \frac{\alpha}{\sqrt{2\pi}}. \tag{16}$$

*Proof.* Since $\Phi'(t) = \varphi(t) = \frac{\mathrm{e}^{-t^2/2}}{\sqrt{2\pi}}$, then

$$\Phi(z+\alpha) - \Phi(z) = \int_z^{z+\alpha} \Phi'(t)\,\mathrm{d}t = \int_z^{z+\alpha} \frac{\mathrm{e}^{-t^2/2}}{\sqrt{2\pi}}\,\mathrm{d}t \leq \int_z^{z+\alpha} \frac{1}{\sqrt{2\pi}}\,\mathrm{d}t = \frac{\alpha}{\sqrt{2\pi}}. \qquad \square$$

**Lemma 4** (a Gaussian upper-tail gap estimate). *For any $z \geq 0$ and any $\alpha > 0$,*

$$\Phi(z+\alpha) - \Phi(z) \leq \sqrt{2\pi}\,(\Phi(\alpha) - \Phi(0))\varphi(z) \leq \alpha\,\varphi(z). \tag{17}$$

*Proof.* Define a function $h : \mathbb{R}_{\geq 0} \to \mathbb{R}$ as

$$h(z) := \frac{\Phi(z+\alpha) - \Phi(z)}{\varphi(z)}, \qquad z \geq 0. \tag{18}$$

Since $\Phi'(z) = \varphi(z)$, and $\varphi'(z) = -z\varphi(z)$, then

$$h'(z) = \frac{(\varphi(z+\alpha) - \varphi(z))\Phi'(z) + (\Phi(z+\alpha) - \Phi(z))(-\varphi'(z))}{\varphi(z)^2} \tag{19}$$

$$= \frac{(\varphi(z+\alpha) - \varphi(z))\varphi(z) + (\Phi(z+\alpha) - \Phi(z))(z\varphi(z))}{\varphi(z)^2} \tag{20}$$

$$= \frac{\varphi(z+\alpha) - \varphi(z) + z(\Phi(z+\alpha) - \Phi(z))}{\varphi(z)} \tag{21}$$

$$= \frac{\int_z^{z+\alpha} \varphi'(t)\,\mathrm{d}t + z\int_z^{z+\alpha} \Phi'(t)\,\mathrm{d}t}{\varphi(z)} \tag{22}$$

$$= \frac{\int_z^{z+\alpha} (-t\varphi(t))\,\mathrm{d}t + z\int_z^{z+\alpha} \varphi(t)\,\mathrm{d}t}{\varphi(z)} \tag{23}$$

$$= -\frac{\int_z^{z+\alpha} (t - z)\varphi(t)\,\mathrm{d}t}{\varphi(z)} < 0. \tag{24}$$

Hence, $h(z)$ is a decreasing function. It follows from Lemma 3 that

$$\frac{\Phi(z+\alpha) - \Phi(z)}{\varphi(z)} = h(z) \le h(0) = \frac{\Phi(\alpha) - \Phi(0)}{\varphi(0)} = \sqrt{2\pi}\,(\Phi(\alpha) - \Phi(0)) \tag{25}$$

$$= \sqrt{2\pi} \int_0^\alpha \Phi'(t)\,\mathrm{d}t = \sqrt{2\pi} \int_0^\alpha \frac{\mathrm{e}^{-t^2/2}}{\sqrt{2\pi}}\,\mathrm{d}t \le \sqrt{2\pi} \int_0^\alpha \frac{1}{\sqrt{2\pi}}\,\mathrm{d}t = \int_0^\alpha \mathrm{d}t = \alpha. \qquad \square$$

**Lemma 5** (a Gaussian inverse estimate). *For every* $0 < v \le \frac{1}{2}$,

$$\varphi(\Phi^{-1}(1-v)) \le v\sqrt{2\ln\frac{1}{v}}. \tag{26}$$

*Proof.* Let $z := \Phi^{-1}(1-v) \ge 0$, so that $v = 1 - \Phi(z) = \overline{\Phi}(z)$.

Note that it is equivalent to show that

$$\ln\left(\frac{1}{\overline{\Phi}(z)}\right) \ge \frac{\varphi(z)^2}{2\overline{\Phi}(z)^2}. \tag{27}$$

Define a function $h : \mathbb{R}_{\ge 0} \to \mathbb{R}$ as

$$h(z) := \ln\left(\frac{1}{\overline{\Phi}(z)}\right) - \frac{\varphi(z)^2}{2\overline{\Phi}(z)^2}, \qquad z \ge 0. \tag{28}$$

Since $z \ge 0$, then by Gordon (1941),

$$\frac{\varphi(z)}{\overline{\Phi}(z)} \ge z \ge \frac{z}{2}. \tag{29}$$

and by Birnbaum (1942),

$$\frac{\varphi(z)}{\overline{\Phi}(z)} \le \frac{2}{\sqrt{z^2+4}-z} = \frac{\sqrt{z^2+4}+z}{2} = \frac{z}{2} + \frac{\sqrt{z^2+4}}{2}. \tag{30}$$

Together, we have

$$\frac{z}{2} < \frac{\varphi(z)}{\overline{\Phi}(z)} \le \frac{z}{2} + \frac{\sqrt{z^2+4}}{2}. \tag{31}$$

Furthermore, since $\overline{\Phi}'(z) = -\varphi(z)$, and $\varphi'(z) = -z\varphi(z)$,

$$h'(z) = \frac{\varphi(z)}{\overline{\Phi}(z)}\left(1 + z\frac{\varphi(z)}{\overline{\Phi}(z)} - \left(\frac{\varphi(z)}{\overline{\Phi}(z)}\right)^2\right) \tag{32}$$

$$= \frac{\varphi(z)}{\overline{\Phi}(z)}\left(\frac{\varphi(z)}{\overline{\Phi}(z)} - \frac{z}{2} + \frac{\sqrt{z^2+4}}{2}\right)\left(\frac{z}{2} + \frac{\sqrt{z^2+4}}{2} - \frac{\varphi(z)}{\overline{\Phi}(z)}\right) \tag{33}$$

$$\ge 0. \tag{34}$$

Hence, the function $h(z)$ is non-decreasing w.r.t. $z$. It follows that for any $0 < v \le \frac{1}{2}$,

$$\ln\left(\frac{1}{v}\right) - \frac{\varphi(\Phi(1-v))^2}{2v^2} = \ln\left(\frac{1}{\overline{\Phi}(z)}\right) - \frac{\varphi(z)^2}{2\overline{\Phi}(z)^2} = h(z) \ge h(0) = \ln 2 - \frac{1}{\pi} > 0. \tag{35}$$

Therefore, $\varphi(\Phi^{-1}(1-v)) \le v\sqrt{2\ln\frac{1}{v}}$. $\qquad\qquad \square$

**Lemma 6** (a Gamma integral). *Let* $\Gamma(\beta)$ *denote the Gamma function* $(\beta > 0)$. *For any* $\alpha, \beta > 0$,

$$\int_0^1 t^{\alpha-1}\left(\ln\frac{1}{t}\right)^{\beta-1}\mathrm{d}t = \frac{\Gamma(\beta)}{\alpha^\beta}. \tag{36}$$

*In particular,*

$$\int_0^1 t\sqrt{\ln\frac{1}{t}}\,\mathrm{d}t = \frac{\Gamma\left(\frac{1}{2}+1\right)}{(1+1)^{1/2+1}} = \frac{\sqrt{\pi}}{4\sqrt{2}}. \tag{37}$$

*Proof.* Let $z := \alpha \ln \frac{1}{t}$. Then,

$$\int_0^1 t^{\alpha-1} \left( \ln \frac{1}{t} \right)^{\beta-1} \mathrm{d}t = \int_0^{+\infty} \left( \mathrm{e}^{-z/\alpha} \right)^{\alpha-1} \left( \frac{z}{\alpha} \right)^{\beta-1} \frac{\mathrm{e}^{-z/\alpha}}{\alpha} \, \mathrm{d}z \tag{38}$$

$$= \frac{1}{\alpha^\beta} \int_0^{+\infty} \mathrm{e}^{-z} z^{\beta-1} \, \mathrm{d}z = \frac{1}{\alpha^\beta} \Gamma(\beta). \qquad \square$$

**Lemma 7** (a mixed integral estimate). *For every $\alpha > 0$ and $0 < \beta \le \alpha$,*

$$\int_0^\beta t \mathrm{e}^{-t} \sqrt{\ln \frac{\alpha}{t}} \, \mathrm{d}t \le \sqrt{\ln \alpha} \left( 1 - \mathrm{e}^{-\beta+\ln(\beta+1)} \right) + \frac{\sqrt{\pi}}{4\sqrt{2}}. \tag{39}$$

*Proof.* Since $\ln \frac{1}{t} \ge 0$ only when $0 \le t \le 1$, then by the triangle inequality,

$$\sqrt{\ln \frac{\alpha}{t}} = \sqrt{\ln \alpha + \ln \frac{1}{t}} \le \sqrt{\ln \alpha + \mathbb{1}_{[0 \le t \le 1]} \ln \frac{1}{t}} \tag{40}$$

$$\le \sqrt{\ln \alpha} + \sqrt{\mathbb{1}_{[0 \le t \le 1]} \ln \frac{1}{t}} = \sqrt{\ln \alpha} + \mathbb{1}_{[0 \le t \le 1]} \sqrt{\ln \frac{1}{t}}. \tag{41}$$

It follows from Lemma 6 that

$$\int_0^\beta t \mathrm{e}^{-t} \sqrt{\ln \frac{\alpha}{t}} \, \mathrm{d}t \le \int_0^\beta t \mathrm{e}^{-t} \left( \sqrt{\ln \alpha} + \mathbb{1}_{[0 \le t \le 1]} \sqrt{\ln \frac{1}{t}} \right) \mathrm{d}t \tag{42}$$

$$= \sqrt{\ln \alpha} \int_0^\beta t \mathrm{e}^{-t} \, \mathrm{d}t + \int_0^{\min\{1,\beta\}} t \mathrm{e}^{-t} \sqrt{\ln \frac{1}{t}} \, \mathrm{d}t \tag{43}$$

$$\le \sqrt{\ln \alpha} \int_0^\beta t \mathrm{e}^{-t} \, \mathrm{d}t + \int_0^1 t \mathrm{e}^{-t} \sqrt{\ln \frac{1}{t}} \, \mathrm{d}t \tag{44}$$

$$\le \sqrt{\ln \alpha} \int_0^\beta t \mathrm{e}^{-t} \, \mathrm{d}t + \int_0^1 t \sqrt{\ln \frac{1}{t}} \, \mathrm{d}t \tag{45}$$

$$= \sqrt{\ln \alpha} \left( 1 - \mathrm{e}^{-\beta+\ln(\beta+1)} \right) + \frac{\sqrt{\pi}}{4\sqrt{2}}. \qquad \square$$

**Lemma 8** (an integer function bound). *For every integer $n \ge 3$,*

$$\frac{\sqrt{2}}{(n-2)^2} \left( \sqrt{\ln(n-2)}(1 - \mathrm{e}^{-\frac{n}{2}-1+\ln \frac{n}{2}}) + \frac{\sqrt{\pi}}{4\sqrt{2}} \right) \le \frac{3}{2} \sqrt{\frac{\pi}{\ln 3}} \frac{\sqrt{\ln n}}{n(n-1)}. \tag{46}$$

*Proof.* Define a function $h : \mathbb{N}_{\ge 3} \to \mathbb{R}$:

$$h(n) := \frac{\frac{\sqrt{2}}{(n-2)^2} \left( \sqrt{\ln(n-2)}(1 - \mathrm{e}^{-\frac{n}{2}-1+\ln \frac{n}{2}}) + \frac{\sqrt{\pi}}{4\sqrt{2}} \right)}{\frac{\sqrt{\ln n}}{n(n-1)}}, \qquad n \ge 3. \tag{47}$$

Note that when $n \ge 9$, we have

$$h(n) \le \frac{\frac{\sqrt{2}}{(n-2)^2} \left( \sqrt{\ln(n-2)} + \frac{\sqrt{\pi}}{4\sqrt{2}} \right)}{\frac{\sqrt{\ln n}}{n(n-1)}} = \frac{\frac{\sqrt{2}}{(n-2)^2} \left( \sqrt{\ln(n-2)} + \frac{\sqrt{\pi}}{4\sqrt{2}} \right)}{\frac{\sqrt{\ln n}}{n(n-1)}} \tag{48}$$

$$= \left( \sqrt{2} \sqrt{\frac{\ln(n-2)}{\ln n}} + \frac{1}{4} \sqrt{\frac{\pi}{\ln n}} \right) \left( 1 + \frac{2}{n-2} + \frac{3}{(n-2)^2} \right) \tag{49}$$

$$\le \left( \sqrt{2} + \frac{1}{4} \sqrt{\frac{\pi}{\ln n}} \right) \left( 1 + \frac{2}{n-2} + \frac{3}{(n-2)^2} \right) \tag{50}$$

$$\le \left( \sqrt{2} + \frac{1}{4} \sqrt{\frac{\pi}{\ln 9}} \right) \left( 1 + \frac{2}{9-2} + \frac{3}{(9-2)^2} \right) \tag{51}$$

$$= \left( \sqrt{2} + \frac{1}{4} \sqrt{\frac{\pi}{\ln 9}} \right) \frac{72}{49} < 2.52. \tag{52}$$

It follows that

$$\frac{\frac{\sqrt{2}}{(n-2)^2}\left(\sqrt{\ln(n-2)} + \frac{\sqrt{\pi}}{4\sqrt{2}}\right)}{\frac{\sqrt{\ln n}}{n(n-1)}} = h(n) \tag{53}$$

$$\leq \max\left\{h(3), h(4), \ldots, h(8), \left(\sqrt{2} + \frac{1}{4}\sqrt{\frac{\pi}{\ln 9}}\right)\frac{72}{49}\right\} \tag{54}$$

$$= h(3) = \frac{3}{2}\sqrt{\frac{\pi}{\ln 3}} < 2.54. \qquad \square \tag{55}$$

**Lemma 9** (a Gaussian integral estimate). *For every integer $n \geq 3$,*

$$\int_0^{+\infty} \Phi(z)^{n-2}\varphi(z)^2\,\mathrm{d}z \leq \frac{3}{2}\sqrt{\frac{\pi}{\ln 3}}\frac{\sqrt{\ln n}}{n(n-1)}. \tag{55}$$

*Proof.* Let $v := 1 - \Phi(z)$ and $t := (n-2)v$. By the fact that $1 - v \leq \mathrm{e}^{-v}$ and Lemmas 5, 7, & 8,

$$\int_0^{+\infty}\Phi(z)^{n-2}\varphi(z)^2\,\mathrm{d}z = \int_0^{1/2}(1-v)^{n-2}\varphi(\Phi^{-1}(1-v))\,\mathrm{d}v \leq \int_0^{1/2}(\mathrm{e}^{-v})^{n-2}\varphi(\Phi^{-1}(1-v))\,\mathrm{d}v \tag{56}$$

$$\leq \int_0^{1/2}(\mathrm{e}^{-v})^{n-2}v\sqrt{2\ln\frac{1}{v}}\,\mathrm{d}v = \frac{\sqrt{2}}{(n-2)^2}\int_0^{\frac{n}{2}-1}t\mathrm{e}^{-t}\sqrt{\ln\frac{n-2}{t}}\,\mathrm{d}t \tag{57}$$

$$\leq \frac{\sqrt{2}}{(n-2)^2}\left(\sqrt{\ln(n-2)}(1-\mathrm{e}^{-\frac{n}{2}-1+\ln\frac{n}{2}}) + \frac{\sqrt{\pi}}{4\sqrt{2}}\right) \tag{58}$$

$$\leq \frac{3}{2}\sqrt{\frac{\pi}{\ln 3}}\frac{\sqrt{\ln n}}{n(n-1)} < 2.54\frac{\sqrt{\ln n}}{n(n-1)}. \qquad \square$$

With the technical lemmata above, we are now ready to prove Theorem 1.

*Proof of Theorem 1.* Let $\boldsymbol{\xi} := \boldsymbol{P}^{(l)}\boldsymbol{x}^{(l)}$ denote the logits of routing weights, so that $\boldsymbol{\pi}^{(l)} = \mathrm{softmax}(\boldsymbol{\xi})$. Let $\xi_{(1)} \geq \cdots \geq \xi_{(n)}$ denote the order statistics of $\boldsymbol{\xi}$ (i.e., $\xi_{(1)}$ is the largest entry of $\boldsymbol{\xi}$, $\xi_{(2)}$ is the second largest entry of $\boldsymbol{\xi}$, etc.). Note that

$$\mathrm{ESS}(\boldsymbol{\pi}^{(l)}) = \frac{\|\boldsymbol{\pi}^{(l)}\|_1^2}{\|\boldsymbol{\pi}^{(l)}\|_2^2} = \frac{\left(\sum_{i=1}^n \pi_i^{(l)}\right)^2}{\sum_{i=1}^n (\pi_i^{(l)})^2} = \frac{\left(\sum_{i=1}^n \mathrm{softmax}(\boldsymbol{\xi})_i\right)^2}{\sum_{i=1}^n \mathrm{softmax}(\boldsymbol{\xi})_i^2} \tag{59}$$

$$= \frac{\left(\sum_{i=1}^n \mathrm{e}^{\xi_i}\right)^2}{\sum_{i=1}^n (\mathrm{e}^{\xi_i})^2} = \frac{\left(\sum_{i=1}^n \mathrm{e}^{\xi_{(i)}}\right)^2}{\sum_{i=1}^n (\mathrm{e}^{\xi_{(i)}})^2} \leq \frac{\left(\sum_{i=1}^n \mathrm{e}^{\xi_{(i)}}\right)^2}{(\mathrm{e}^{\xi_{(1)}})^2} \tag{60}$$

$$= \left(1 + \sum_{i=2}^n \frac{1}{\mathrm{e}^{\xi_{(1)} - \xi_{(i)}}}\right)^2 \leq \left(1 + \sum_{i=2}^n \frac{1}{\mathrm{e}^{\xi_{(1)} - \xi_{(2)}}}\right)^2 \tag{61}$$

$$= \left(1 + \frac{n-1}{\mathrm{e}^{\xi_{(1)} - \xi_{(2)}}}\right)^2 = \left(1 + \frac{1}{\mathrm{e}^{\xi_{(1)} - \xi_{(2)} - \ln(n-1)}}\right)^2. \tag{62}$$

Since $\boldsymbol{P}^{(l)}$ have i.i.d. $\mathcal{N}(0, \sigma^2)$ entries, then $\boldsymbol{\xi} = \boldsymbol{P}^{(l)}\boldsymbol{x}^{(l)}$ have i.i.d $\mathcal{N}(0, \sigma^2\|\boldsymbol{x}^{(l)}\|_2^2)$ entries. Let

$$\kappa := \frac{1}{\frac{3}{2}\sqrt{\frac{\pi}{\ln 3}}\ln n + \frac{1}{\sqrt{2\pi}}2^{n-\log_2 n-1}}. \tag{63}$$

For any $0 < \delta < 1$, with $z_{(i)} := \frac{\xi_{(i)} - 0}{\sigma \|\boldsymbol{x}^{(l)}\|_2}$ $(i = 1, 2)$, by Lemmas 3, 4, & 9,

$$\mathbb{P}[\xi_{(1)} - \xi_{(2)} \leq \delta \kappa \sigma \|\boldsymbol{x}^{(l)}\|_2] = \mathbb{P}[z_{(1)} - z_{(2)} \leq \delta \kappa] \tag{64}$$

$$= \int_{-\infty}^{+\infty} \int_{z_{(2)}}^{z_{(2)} + \delta \kappa} n(n-1) \varphi(z_{(1)}) \varphi(z_{(2)}) \Phi(z_{(2)})^{n-2} \, \mathrm{d}z_{(1)} \, \mathrm{d}z_{(2)} \tag{65}$$

$$= n(n-1) \int_{-\infty}^{+\infty} \int_{z_{(2)}}^{z_{(2)} + \delta \kappa} \varphi(z_{(1)}) \, \mathrm{d}z_{(1)} \, \varphi(z_{(2)}) \Phi(z_{(2)})^{n-2} \, \mathrm{d}z_{(2)} \tag{66}$$

$$= n(n-1) \int_{-\infty}^{+\infty} (\Phi(z_{(2)} + \delta \kappa) - \Phi(z_{(2)})) \varphi(z_{(2)}) \Phi(z_{(2)})^{n-2} \, \mathrm{d}z_{(2)} \tag{67}$$

$$= n(n-1) \left( \int_{-\infty}^{0} + \int_{0}^{+\infty} \right) (\Phi(z_{(2)} + \delta \kappa) - \Phi(z_{(2)})) \varphi(z_{(2)}) \Phi(z_{(2)})^{n-2} \, \mathrm{d}z_{(2)} \tag{68}$$

$$\leq n(n-1) \left( \int_{-\infty}^{0} \frac{\delta \kappa}{\sqrt{2\pi}} \varphi(z_{(2)}) \Phi(z_{(2)})^{n-2} \, \mathrm{d}z_{(2)} + \int_{0}^{+\infty} \delta \kappa \varphi(z_{(2)}) \varphi(z_{(2)}) \Phi(z_{(2)})^{n-2} \, \mathrm{d}z_{(2)} \right) \tag{69}$$

$$= \delta \kappa n(n-1) \left( \frac{1}{\sqrt{2\pi}} \int_{-\infty}^{0} \varphi(z_{(2)}) \Phi(z_{(2)})^{n-2} \, \mathrm{d}z_{(2)} + \int_{0}^{+\infty} \Phi(z_{(2)})^{n-2} \varphi(z_{(2)})^2 \, \mathrm{d}z_{(2)} \right) \tag{70}$$

$$= \delta \kappa n(n-1) \left( \frac{1}{\sqrt{2\pi}} \frac{\Phi(0)^{n-1} - \Phi(-\infty)^{n-1}}{n-1} + \int_{0}^{+\infty} \Phi(z_{(2)})^{n-2} \varphi(z_{(2)})^2 \, \mathrm{d}z_{(2)} \right) \tag{71}$$

$$= \delta \kappa n(n-1) \left( \frac{1}{\sqrt{2\pi}(n-1)2^{n-1}} + \int_{0}^{+\infty} \Phi(z_{(2)})^{n-2} \varphi(z_{(2)})^2 \, \mathrm{d}z_{(2)} \right) \tag{72}$$

$$\leq \delta \kappa n(n-1) \left( \frac{1}{\sqrt{2\pi}(n-1)2^{n-1}} + \frac{3}{2} \sqrt{\frac{\pi}{\ln 3}} \frac{\sqrt{\ln n}}{n(n-1)} \right) \tag{73}$$

$$= \delta \kappa \left( \frac{3}{2} \sqrt{\frac{\pi}{\ln 3}} \sqrt{\ln n} + \frac{n}{\sqrt{2\pi} 2^{n-1}} \right) = \delta \kappa \left( \frac{3}{2} \sqrt{\frac{\pi}{\ln 3}} \sqrt{\ln n} + \frac{1}{\sqrt{2\pi} 2^{n - \log_2 n - 1}} \right) = \delta. \tag{74}$$

This implies $\mathbb{P}[\xi_{(1)} - \xi_{(2)} > \delta \kappa \sigma \|\boldsymbol{x}^{(l)}\|_2] \geq 1 - \delta$. It follows that with probability at least $1 - \delta$,

$$\mathrm{ESS}(\boldsymbol{\pi}^{(l)}) \leq \left( 1 + \frac{1}{e^{\xi_{(1)} - \xi_{(2)} - \ln(n-1)}} \right)^2 \leq \left( 1 + \frac{1}{e^{\delta \kappa \sigma \|\boldsymbol{x}^{(l)}\|_2 - \ln(n-1)}} \right)^2 \tag{75}$$

$$= \left( 1 + \frac{1}{\exp \left( \frac{\delta \sigma \|\boldsymbol{x}^{(l)}\|_2}{\frac{3}{2} \sqrt{\frac{\pi}{\ln 3}} \sqrt{\ln n} + \frac{1}{\sqrt{2\pi} 2^{n - \log_2 n - 1}}} - \ln(n-1) \right)} \right)^2. \qquad \square$$

## A.2 PROOF OF THEOREM 2

Before stating our proof of Theorem 1, we present a technical lemma that we will employ.

To simplify notation, we omit the superscript $^{(l)}$ in this proof. For an ordered subset $\mathcal{I} = (i_1, \ldots, i_k) \subseteq \{1, \ldots, n\}$, let $q(\mathcal{I})$ denote the probability of sampling an ordered subset $\mathcal{I}$ from $\boldsymbol{q}$ without replace:

$$Q(\mathcal{I}) = Q(i_1, \ldots, i_k) := \prod_{j=1}^{k} \frac{q_{i_j}}{1 - \sum_{j'=1}^{j-1} q_{i_{j'}}}. \tag{76}$$

Let $\mathcal{P}_k$ denote the set of permutations over $\{1, \ldots, n\}$. For $\varpi \in \mathcal{P}_k$, define the permutation action as $\varpi(i_1, \ldots, i_k) := (i_{\varpi(1)}, \ldots, i_{\varpi(k)})$. Let $\overline{Q}(\mathcal{I})$ denote the probability of sampling an unordered subset $\mathcal{I}$ from $\boldsymbol{q}$ without replacement:

$$\overline{Q}(\mathcal{I}) = \mathbb{P}_{\mathcal{I} \sim \boldsymbol{q}}[\mathcal{I}] = \sum_{\varpi \in \mathcal{P}_n} Q(\varpi(\mathcal{I})). \tag{77}$$

**Lemma 10** (swapping a pair). *Given a size-$k$ subset $\mathcal{I} \subseteq \{1, \ldots, n\}$, for a LoRA $i \in \mathcal{I}$ and another LoRA $i^\dagger \in \{1, \ldots, n\} \setminus \mathcal{I}$, if $q_i \leq q_{i^\dagger}$, then replacing $i$ with $i^\dagger$ increases the unordered sampling probability:*

$$\overline{Q}((\mathcal{I} \setminus \{i\}) \cup \{i^\dagger\}) > \overline{Q}(\mathcal{I}). \tag{78}$$

*Proof.* Say $\mathcal{I} = (i_1, \ldots, i_k)$. Without loss of generality, say $i_1 = i$, and let $\mathcal{I}^\dagger := (i^\dagger, i_2, \ldots, i_k)$ denote the ordered subset after replacing $i$ with $i^\dagger$. For any permutation $\varpi \in \mathcal{P}_k$, let $j_\varpi := \varpi^{-1}(1)$ denote the order of $i$ under permutation $\varpi$ (i.e., $\varpi(\mathcal{I})_{j_\varpi} = i$). Since $q_i \leq q_{i^\dagger}$, then

$$\frac{Q(\varpi(\mathcal{I}^\dagger))}{Q(\varpi(\mathcal{I}))} = \frac{q_{i^\dagger}}{q_i} \prod_{j=j_\varpi+1}^{k} \frac{1 - \sum_{j'=1}^{j-1} q_{i_{j'}}}{1 - q_{i^\dagger} + q_i - \sum_{j'=1}^{j-1} q_{i_{j'}}} \tag{79}$$

$$= \frac{q_{i^\dagger}}{q_i} \prod_{j=j_\varpi+1}^{k} \frac{1}{1 - \frac{q_{i^\dagger} - q_i}{1 - \sum_{j'=1}^{j-1} q_{i_{j'}}}} \tag{80}$$

$$\geq \frac{q_{i^\dagger}}{q_i} \prod_{j=j_\varpi+1}^{k} 1 = \frac{q_{i^\dagger}}{q_i} \geq 1. \tag{81}$$

This means $Q(\varpi(\mathcal{I}^\dagger)) \geq Q(\varpi(\mathcal{I}))$. It follows that

$$\overline{Q}((\mathcal{I} \setminus \{i\}) \cup \{i^\dagger\}) = \overline{Q}(\mathcal{I}^\dagger) = \sum_{\varpi \in \mathcal{P}_n} Q(\varpi(\mathcal{I}^\dagger)) \tag{82}$$

$$\geq \sum_{\varpi \in \mathcal{P}_n} Q(\varpi(\mathcal{I})) = \overline{Q}(\mathcal{I}). \qquad \square$$

We are now ready to prove Theorem 2.

*Proof of Theorem 2.* Suppose that

$$\mathcal{I}^\dagger := \operatorname*{argtop}_{i=1}^{n} {}_k q_i \neq \mathcal{I}^*, \tag{83}$$

where we break ties arbitrarily. We will show that this premise leads to a contradiction.

Recall that by definition,

$$\overline{Q}(\mathcal{I}^*) = \mathbb{P}_{\mathcal{I} \sim \boldsymbol{q}}[\mathcal{I} = \mathcal{I}^*] > \frac{1}{2}. \tag{84}$$

Since $\mathcal{I}^\dagger \neq \mathcal{I}^*$, then $k^\cap := |\mathcal{I}^* \cap \mathcal{I}^\dagger| < k$. Say $\mathcal{I}^* \setminus \mathcal{I}^\dagger = \{i_1^*, \ldots, i_{k-k^\cap}^*\}$, $\mathcal{I}^\dagger \setminus \mathcal{I}^* = \{i_1^\dagger, \ldots, i_{k-k^\cap}^\dagger\}$. Construct a series of subsets inductively as follows. Define $\widetilde{\mathcal{I}}_0 := \mathcal{I}^*$. For $j = 1, \ldots, k - k^\cap$, define $\widetilde{\mathcal{I}}_j$ by replacing $i_j^*$ from $\widetilde{\mathcal{I}}_{j-1}$ with $i_j^\dagger$ and inheriting all other LoRAs from $\widetilde{\mathcal{I}}_{j-1}$. Finally, we have $\widetilde{\mathcal{I}}_{k-k^\cap} = \mathcal{I}^\dagger$. Since $\mathcal{I}^\dagger$ consists of LoRAs $i$ with top-$k$ $q_i$, then $q_{i_j^*} \leq q_{i_j^\dagger}$ for all $j = 1, \ldots, k - k^\cap$. Hence, by Lemma 10, $\overline{Q}(\widetilde{\mathcal{I}}_j) \geq \overline{Q}(\widetilde{\mathcal{I}}_{j-1})$ for all $j = 1, \ldots, k - k^\cap$. Together,

$$\overline{Q}(\mathcal{I}^\dagger) = \overline{Q}(\widetilde{\mathcal{I}}_{k-k^\cap}) \geq \overline{Q}(\widetilde{\mathcal{I}}_{k-k^\cap-1}) \geq \cdots \geq \overline{Q}(\widetilde{\mathcal{I}}_0) = \overline{Q}(\mathcal{I}^*) > \frac{1}{2}. \tag{85}$$

It follows that

$$\overline{Q}(\mathcal{I}^\dagger) + \overline{Q}(\mathcal{I}^*) > \frac{1}{2} + \frac{1}{2} = 1. \tag{86}$$

However, this contradicts the fact that

$$\overline{Q}(\mathcal{I}^\dagger) + \overline{Q}(\mathcal{I}^*) \leq \sum_{\mathcal{I}} \overline{Q}(\mathcal{I}) = 1, \tag{87}$$

falsifying the premise. Therefore,

$$\operatorname*{argtop}_{i=1}^{n} {}_k q_i = \mathcal{I}^*. \qquad \square$$

## B   RELATED WORK (CONT'D)

PEFT approaches can be broadly categorized into four groups: prompt tuning, prefix tuning, adapter-based methods, and low-rank adaptation methods. Early methods such as prompt tuning (Liu et al., 2021a; Shi & Lipani, 2023; Lester et al., 2021; Zang et al., 2022; Wang et al., 2022) and prefix tuning (Li & Liang, 2021; Le et al., 2024; Chen et al., 2022; Petrov et al., 2023) introduce small continuous prompts, but often struggle to scale to deeper layers or larger models due to limited expressivity. Adapter-based methods (He et al., 2022; Rücklé et al., 2020; Jie et al., 2023) mitigate some of these issues by inserting lightweight bottleneck modules into transformer layers. However, as the depth and dimensionality of models increase, the parameter overhead of adapters can become substantial, creating significant bottlenecks in computation and scalability. To address these limitations, low-rank adaptation methods (Hu et al., 2022; Valipour et al., 2022; Zhang et al., 2023; Yang et al., 2023) are proposed. These methods inject rank-constrained updates into weight matrices, striking a favorable balance between expressivity and parameter cost, and have become a de facto standard for many adaptation tasks. Specifically, LoRA (Hu et al., 2022) introduces two trainable low-rank matrices while keeping the original model weights frozen. By training these matrices to approximate parameter perturbations, LoRA achieves effective fine-tuning with minimal overhead. Building on this idea, DyLoRA (Valipour et al., 2022) dynamically trains LoRA modules across a range of ranks within a predefined budget rather than fixing the rank. AdaLoRA (Zhang et al., 2023) reformulates parameter perturbations using singular value decomposition (SVD), fine-tuning across the three SVD components for improved flexibility. Laplace-LoRA (Yang et al., 2023) takes a Bayesian perspective, applying a post-hoc Laplace approximation to the posterior distribution over LoRA parameters, thereby offering a principled uncertainty-aware extension.

## C   USE OF LLMS

We have used multiple LLMs (including ChatGPT, Gemini, Claude, and Llama) to refine paper writing and to draft the LATEX code of mathematical equations.

