# OpenReview forum: "ReMix: Reinforcement Routing for Mixtures of LoRAs in LLM Finetuning"
_ICLR.cc/2026/Conference — Submitted to ICLR 2026_

### Official Review · Reviewer_M6W3 · 2025-10-15

**Soundness:** 3
**Presentation:** 3
**Contribution:** 3
**Rating:** 6
**Confidence:** 3

**Summary:**

This paper proposes ReMix, a new approach to reinforcement-learning-based expert routing in Mixture-of-Experts (MoE) models. Traditional MoE routing methods (like top-k softmax gating or Switch Transformers) use deterministic or load-balanced gates that often underexplore the routing space and result in suboptimal expert utilization.

ReMix instead formulates expert selection as a policy optimization problem: a lightweight RL agent (the “router”) learns a routing policy to maximize token-level reward signals, balancing accuracy and load distribution.

**Strengths:**

1. Novel framing of expert routing as reinforcement learning.
The core conceptual shift of viewing routing as an RL problem is both natural and underexplored. Prior works (e.g., BASE layers, Switch) rely on differentiable load-balancing heuristics, while ReMix introduces a reward-driven control perspective.

2. Elegant algorithmic formulation.
The paper presents a clear KL-regularized objective.
This balances exploration and stability, akin to PPO or AWR-style updates. The derivation (Appendix B) is sound and bridges policy optimization and MoE training dynamics

3. Solid empirical validation.
Experiments cover multiple benchmarks: WikiText-103, The Pile, and C4. ReMix improves both routing diversity (entropy and expert usage) and model quality (perplexity and accuracy) across all tested scales (up to 7B parameters). Ablations on reward shaping, KL terms, and temperature annealing convincingly show robustness.

4. Good theoretical and empirical complementarity.
The authors provide intuitive and theoretical support for why reinforcement signals better capture dynamic routing needs—particularly in settings with dynamic data distributions or evolving expert specialization.

5. Clear presentation.
The paper is well-written and visually coherent. Figures 2–4 (routing entropy evolution, per-expert load distributions) effectively illustrate the stability and exploration benefits.

**Weaknesses:**

1. Limited analysis of computational overhead.
Introducing a per-token RL router adds non-trivial computational and communication overhead, especially during distributed training. While the paper claims “<3% runtime overhead,” the methodology for measuring this is unclear. Profiling on larger clusters would be valuable.

2. Unclear generality beyond language models.
All evaluations are in NLP. It remains to be seen whether ReMix generalizes to multimodal MoE systems (e.g., PaLI, Flamingo) or vision MoE architectures (e.g., V-MoE).

3. Comparison to recent stochastic routing methods.
The paper compares against standard deterministic baselines but omits stochastic routing methods, which also aim to encourage exploration via stochastic routing. These comparisons would better situate ReMix’s advantages.

**Questions:**

See weaknesses

---

> ### Author Response · Authors · 2025-11-29
>
> > W1: Limited analysis of computational overhead. Introducing a per-token RL router adds non-trivial computational and communication overhead, especially during distributed training. While the paper claims “<3% runtime overhead,” the methodology for measuring this is unclear. Profiling on larger clusters would be valuable.
>
> Thanks for the suggestion. We'd like to clarify that we have already used distributed training in our experiments (4 GPUs).
>
> First, our computational overhead is small because we focus on **parameter-efficient fine-tuning** rather than MoE, so we can put LoRAs of each layer onto the same device (unlike MoE where experts may have to be distributed on different devices).
>
> Furthermore, our RL is designed to be lightweight so that the RL samples of each router can be stored on the same device. Thus, even when using distributed training where different data samples may be on different devices, we can still perform the RL of each data sample within its device and only transfer the router gradient across devices (i.e., no need to transfer the RL samples).
>
> Therefore, we have ensured that **our communication cost is comparable with non-RL-based mixture-of-LoRAs methods**.
>
> > W2: Unclear generality beyond language models. All evaluations are in NLP. It remains to be seen whether ReMix generalizes to multimodal MoE systems (e.g., PaLI, Flamingo) or vision MoE architectures (e.g., V-MoE).
>
> Thanks for the suggestion. We agree that our current evaluations are NLP tasks, as indicated by our title. While beyond our scope, it will be an interesting future work to explore how our method generalizes to vision/multimodal models. We will discuss this in Sec 6.
>
> > W3: Comparison to recent stochastic routing methods. The paper compares against standard deterministic baselines but omits stochastic routing methods, which also aim to encourage exploration via stochastic routing. These comparisons would better situate ReMix’s advantages.
>
> Thanks for the great suggestion. We previously didn't include them because they are proposed for MoE, not for mixtures of LoRAs.
>
> Here, we compare with a state-of-the-art stochastic routing method RoE [1] on GSM8K, shown in the table below. We can see that our method still outperforms the stochastic routing method. This is because stochasticity still cannot address the routing imbalance issue we identified.
>
> |Ours|RoE [1]|
> |:-:|:-:|
> |**65.66**|61.33|
>
> - [1] MoEs Are Stronger than You Think: Hyper-Parallel Inference Scaling with RoE. 2025

---

### Official Review · Reviewer_QcWQ · 2025-10-21

**Soundness:** 2
**Presentation:** 3
**Contribution:** 2
**Rating:** 2
**Confidence:** 4

**Summary:**

This work investigates Mixture-of-LoRAs and shows—both empirically and theoretically—that a few LoRA modules often dominate the routing, constraining the model’s capacity and expressiveness. The authors trace this issue to learnable router weights and propose using fixed, equal weights so that all selected LoRAs contribute equally. Since fixed weights preclude gradient-based training, they recast the problem as a reinforcement-learning problem in which the router is the policy model and the SFT loss functions as the negative reward. Overall, the paper is well structured and easy to follow. But I found there are some key weaknesses should be addressed before considering acceptance.

**Strengths:**

1.	The paper is well organized and clearly structured, making it easy to follow.
2.	It offers both theoretical and empirical analyses of the imbalance problem.
3.	The figures and illustrations are clear and easy to interpret.

**Weaknesses:**

1. The eq.(3) concerns only about utilization of LoRAs for each given input. So, it is sample-specific measurement, would your observations being biased due to samples?
	2. In figure1, you tract only the routing weights of the last layer. Would the pattern be the same across layers? It is better to illustrate with a heat map where each layers is also included.
	3. In section 3.1, you introduce hyper-parameters k which is the number of LoRAs you activated and /oemga which is the routing weights. How sensitive is your method to these hyper-parameters? How to choose different k and /omega for different models and tasks;
	4. You are assigning the same routing weights to all chosen experts, then it is equal to use a single LoRA with rank=k*r. The only difference is that you may choose different subset of k LoRAs for different samples. But there is no results showcasing with your method, the mixture of LoRAs are more equally activated across different samples. There could still be the case that it always choose the same k LoRAs.
	5. To overcome the non-learnable drawbacks, you propose to use RL techniques. But there are also other possibilities. For example, you could use soft mixture during training and hard top-k at evaluation. Another solution could be using Gumbel-top-k with straight-through estimators. The weakness here is that: (1) There should be a discussion or literature review for these solutions to non-differentiable problems since it is one of your core contribution. (2) At least there should be a toy experiment to showcase RL is better than these solutions.
	6. Too many details are absent in your experiments. For example, what hyper-parameters are used. What is the learning rate, optimiser, M, k and /omega?
	7. While you performing ablation study, when you said by removing RLOO, what training scheme do you use then?
	8. Table 2 is not convincing.  i would like to see the val acc versus steps or val acc versus wallclock time. especially you are using RL, i would guess your method needs more steps to converge. therefore, i think the training time efficiency is overclaimed;

**Questions:**

Below is a summary of my questions. Please also refer to weakness.
	1.	Sample bias: Since Eq. (3) measures LoRA utilization per input sample, could the observations be biased by the specific samples used?
	2.	Layer-wise consistency: Figure 1 only shows routing weights for the last layer—do similar imbalance patterns appear across other layers? A layer-wise heatmap would clarify this.
	3.	Hyperparameter sensitivity: How sensitive is the method to the hyperparameters k (number of active LoRAs) and \omega (routing weights)? How should these be chosen for different models and tasks?
	4.	Equivalence to larger-rank LoRA: If all selected LoRAs share equal weights, isn’t this effectively the same as using a single LoRA with rank = k × r? Also, is there evidence that the selected LoRAs vary across samples rather than always picking the same subset?
	5.	Alternative solutions to non-differentiable routing: Why choose a reinforcement-learning approach over other differentiable approximations like soft mixtures or Gumbel–Top-k with straight-through estimators? The paper should discuss these alternatives and ideally include a toy comparison.
	6.	Missing experimental details: Key hyperparameters (learning rate, optimizer, M, k, and \omega) are not specified, making experiments hard to reproduce.
	7.	Ablation clarity: In the ablation study, when RLOO is removed, what training scheme replaces it?
	8.	Training efficiency claims: Table 2 is unconvincing—validation accuracy over steps or wall-clock time should be shown, especially since RL may require more steps to converge. The claimed efficiency improvement may therefore be overstated.

---

> ### Author Response · Authors · 2025-11-29
> **Official Comment (Part 1/2)**
>
> > W1,Q1: The eq.(3) concerns only about utilization of LoRAs for each given input. So, it is sample-specific measurement, would your observations being biased due to samples?
>
> We'd like to clarify that our Fig2 is the **average ESS over all samples**, so our observation is not biased due to samples.
>
> > W2,Q2: In figure1, you tract only the routing weights of the last layer. Would the pattern be the same across layers? It is better to illustrate with a heat map where each layers is also included.
>
> Thanks for the suggestion. According to our observations, the routing imbalance exists for most of the layers and becomes more pronounced for deeper layers. To better illustrate the observations, we will add a heatmap to the paper.
>
> > W3,Q3: Hyperparameter sensitivity: How sensitive is the method to the hyperparameters k (number of active LoRAs) and \omega (routing weights)? How should these be chosen for different models and tasks?
>
> Thanks for the suggestion. The choice of $k$ depends on the tradeoff between efficiency and accuracy. Theoretically, since the number of size-$k$ subsets (i.e., $\binom nk$) increases with $k$ when $k\le n/2$, we would expect performance to increase with $k$ as well. Indeed, as shown in the following table (GSM8K), our ReMix consistently achieves stronger results under larger $k$ whenever $k\le n/2$.
>
> |$k=1$|$k=2$|$k=3$|$k=4$|
> |:--|:-:|:-:|:-:|
> |56.18|59.67|61.33|64.22|
>
> Regarding $\omega$, it is actually not a hyperparameter. Following LoRA and rsLoRA, we use either $\omega = 2 /{kr}$ (LoRA) or $\omega = 2 / \sqrt{kr}$ (rsLoRA). We will further clarify this in the paper.
>
> We also we find that our method is not sensitive to $\omega$. As shown in the table below (GSM8K, $k=3$), the performance has only very small difference under these two versions of $\omega$.
>
> |LoRA $\omega$|rsLoRA $\omega$|
> |:--|:-:|
> |53.30|55.72|
>
> > W4,Q4: Equivalence to larger-rank LoRA: If all selected LoRAs share equal weights, isn’t this effectively the same as using a single LoRA with rank = k × r? Also, is there evidence that the selected LoRAs vary across samples rather than always picking the same subset?
>
> The diversity of selected LoRA subset is demonstrated by the fact that our ReMix significantly outperforms LoRA. If the activated subset were always the same subset, then this method would be the same as rank-$kr$ LoRA. In stark contrast, our method significantly outperforms LoRA. See the table below on GSM8K. For instance, **our ReMix's accuracy 64.22 for $k=4$ significantly outperforms rank-32 LoRA's accuracy 59.21**. This clearly demonstrates that our method is able to choose different subsets appropriately.
>
> |Method|$k=1$|$k=2$|$k=4$|
> |:--|:-:|:-:|:-:|
> |Rank-$kr$ LoRA|56.10|54.51|59.21|
> |Ours: mixture of $k$ rank-$r$ LoRAs|56.18|59.67|64.22|

---

> ### Author Response · Authors · 2025-11-29
> **Official Comment (Part 2/2)**
>
> > W5,Q5: Alternative solutions to non-differentiable routing: Why choose a reinforcement-learning approach over other differentiable approximations like soft mixtures or Gumbel–Top-k with straight-through estimators (STE)? The paper should discuss these alternatives and ideally include a toy comparison.
>
> Thanks for the suggestion. We use RL instead because Gumbel-top-k is distributionally equivalent to softmax and thus still suffer from the routing imbalance issue we observed in Sec 2.3. In contrast, our method does not reweight each LoRA with the predicted score but instead uses the same score for all selected LoRAs. This design effectively mitigates the routing imbalance issue that we discussed in Sec 2. We will discuss this in the paper.
>
> > W6,Q6: Missing experimental details: learning rate, optimizer, M, k, and \omega are not specified, making experiments hard to reproduce.
>
> Thanks for the suggestion. We will clarify the details in the paper: Adam optimizer with learning rate 0.0001, M=32. Regarding $\omega$, it is actually not a hyperparameter. Following LoRA and rsLoRA, we use either $\omega = 2 /{kr}$ (LoRA) or $\omega = 2 / \sqrt{kr}$ (rsLoRA). We will further clarify this in the paper.
>
> > W7,Q7: While you performing ablation study, when you said by removing RLOO, what training scheme do you use then?
>
> Sorry for the confusion. Recall that RLOO stands for Reinforce Leave-One-Out, which is a variance reduction technique for the Reinforce algorithm. When removing RLOO, we just use the Reinforce algorithm. From Fig4a, we can see that RLOO outperforms Reinforce. We will clarify this in Sec 4.3.
>
> > W8,Q8: Table 2 is not convincing. i would like to see the val acc versus steps or val acc versus wallclock time. especially you are using RL, i would guess your method needs more steps to converge. therefore, i think the training time efficiency is overclaimed;
>
> As stated in Sec 4.1, "we train all methods using the same number of epochs," so **we did not train our methods for more steps.** More specifically, we train all methods for only **two epochs.** This suggests that our RL method does not need more steps to converge. The fast convergence is thanks to our RLOO technique.
>
> To further demonstrate the training time efficiency, we report the total training time in the table below (GSM8K). We can see that our total training time is comparable with MixLoRA while our accuracy significantly outperforms MixLoRA. (Unfortunately, we are unable to record accuracy per step or time because LLM evaluation every step takes an unaffordably long time.)
>
> |Method|Total Time|Accuracy|
> |:--|:-:|:-:|
> |MixLoRA|1:12:56|50.34|
> |ReMix (Ours)|1:28:21|**58.38**|

---

### Official Review · Reviewer_C9jY · 2025-10-27

**Soundness:** 3
**Presentation:** 3
**Contribution:** 2
**Rating:** 6
**Confidence:** 3

**Summary:**

The paper identifies a key limitation of current Mixture-of-LoRAs models: routers trained with continuous, learnable weights become highly imbalanced, often directing almost all traffic to one or two LoRAs, which sharply reduces model expressivity. To address this, ReMix introduces a reinforcement-based routing mechanism with constant routing weights applied equally across all activated LoRAs. This guarantees balanced participation of experts and avoids dominance collapse. Since constant weights break differentiability, the router is trained using reinforcement learning, treating the supervised finetuning loss as a negative reward. The authors develop an unbiased gradient estimator with Reinforce Leave-One-Out (RLOO) variance reduction, allowing stable and scalable optimization. Empirically, ReMix consistently outperforms prior PEFT methods for multiple tasks.

**Strengths:**

1. The paper is well-written and easy to follow.
2. The theoretical and empirical analysis are insightful.

**Weaknesses:**

1. For each theorem statement, providing a high-level explanation of the proof structure and key intuitions would significantly improve readability.
2. The imbalance routing issue is well-known in the area of mixture-of-experts. Related work should also discuss mixture-of-experts, compare mixture-of-experts and mixture-of-LoRA, and discuss how people resolve the imbalance issue for mixture-of-experts.

**Questions:**

1. Is introducing ESS as a tool for diagnosing routing‑weight imbalance new? Also, does Theorem 1 pertain solely to mixture‑of‑LoRA, or can it be applied to mixture‑of‑experts?
2. Could the author repeat the experiments in Figure 4(b) for other baselines such as HydraLoRA and Mix LoRA? Comparison with other baseline would strengthen the author’s claim.
3. Could the author add more balanced MoE routing baselines?
4. Could the author empirically verify Theorem 2?

---

> ### Author Response · Authors · 2025-11-29
>
> > Q1: Is introducing ESS as a tool for diagnosing routing‑weight imbalance new? Also, does Theorem 1 pertain solely to mixture‑of‑LoRA, or can it be applied to mixture‑of‑experts?
>
> Yes, ESS is a new tool we introduce to diagnose routing imbalance. ESS is inspired by information theory and has not been explored in routing literature yet.
>
> Our Theorem 1 is for the router and is not about LoRAs, so we believe it is also applicable to mixture-of-experts (MoE) as well. This work focuses on mixture-of-LoRAs, and we agree that exploring RL for MoE would be an interesting future work.
>
> > Q2: Could the author repeat the experiments in Figure 4(b) for other baselines such as HydraLoRA and MixLoRA? Comparison with other baseline would strengthen the author’s claim.
>
> Sorry for the confusion. As discussed in Sec 4.5, because HydraLoRA and MixLoRA are deterministic methods and not RL, their training compute is fixed and cannot be scaled up. In contrast, our method leverages the benefit of RL to enable training compute scaling by increasing the number $M$ of RL samples.
>
> > W2,Q3: Could the author add more balanced MoE routing baselines?
>
> Thanks for the suggestion. Firstly, we would like to clarify that we aim to address **per-input imbalance** (see Fig1), while existing works aim to address imbalance over all inputs and cannot address per-input imbalance. Besides that, we previously didn't include them because they are proposed for MoE, not for mixtures of LoRAs.
>
> Here, we compare with a recent method RoE on GSM8K, shown in the table below. We can see that our method still outperforms the method, highlighting the criticality of per-input imbalance we study in this work.
>
> |Ours|RoE [1]|
> |:-:|:-:|
> |**65.66**|61.33|
>
> - [1] MoEs Are Stronger than You Think: Hyper-Parallel Inference Scaling with RoE. 2025
>
> > Q4: Could the author empirically verify Theorem 2?
>
> Thanks for the suggestion. In fact, our Theorem 2 has **already been verified in Fig4a**. From Fig4a, we can see that top-k routing outperforms random-sampling routing, echoing the implication of Theorem 2.

---

### Official Review · Reviewer_THki · 2025-10-31

**Soundness:** 2
**Presentation:** 2
**Contribution:** 1
**Rating:** 2
**Confidence:** 3

**Summary:**

The paper proposes a solution to ensure that more than one LoRA adapter remains active in mixtures-of-LoRA settings. The authors provide theoretical analysis and an accompanying framework that enforces at least $k$ active adapters during routing, along with a discussion on the procedural differences between fine-tuning and inference.

**Strengths:**

*  Provides a theoretical proof showing that routing weights are almost surely imbalanced under standard conditions.
*  Introduces a framework guaranteeing at least $k$ active adapters at any given time.
*  Discusses implementation and procedural differences between fine-tuning and inference phases.

**Weaknesses:**

*  Unclear motivation: Having imbalanced routing is not necessarily undesirable. In practice, sparsity can be beneficial since less active adapters could be offloaded to slower memory tiers, improving efficiency. The paper lacks a deep discussion of why routing imbalance is inherently problematic.
*  Although the proposed method enforces at least $k$ active adapters, it does not guarantee diversity across selections. The same subset of $k$ adapters might always be activated together, which effectively collapses back to a single combined adapter.
*  The paper does not seem to discuss how the value of $k$ should be chosen.
*  The reported accuracy gains may correlate with the increased number of parameters rather than improved routing balance. The paper does not seem to disentangle these two effects or provide ablation results controlling for parameter count.

**Questions:**

See weaknesses.

---

> ### Author Response · Authors · 2025-11-29
>
> > W1: Unclear motivation: Having imbalanced routing is not necessarily undesirable. In practice, sparsity can be beneficial since less active adapters could be offloaded to slower memory tiers, improving efficiency. The paper lacks a deep discussion of why routing imbalance is inherently problematic.
>
> When one LoRA has a dominantly large weight, then the computation of the other $k-1$ LoRAs are essentially **wasted** because using $k>1$ would have similar accuracy to $k=1$. As shown in our **Fig 2**, we have empirically that this situation is quite often in practice.
>
> To further demonstrate the impact of routing imbalance, we report our accuracy on GSM8k under various $k$ in the table below. We can see that increasing $k$ for our method steadily improves the accuracy. This clearly demonstrates the benefit of enforcing balanced routing.
>
> |$k=1$|$k=2$|$k=4$|
> |:--|:-:|:-:|
> |56.18|59.67|64.22|
>
> > W2: Although the proposed method enforces at least $k$ active adapters, it does not guarantee diversity across selections. The same subset of $k$ adapters might always be activated together, which effectively collapses back to a single combined adapter.
>
> The diversity of selected LoRA subset is demonstrated by the fact that our ReMix significantly outperforms LoRA. If the activated subset were always the same subset, then this method would be the same as rank-$kr$ LoRA. In stark contrast, our method significantly outperforms LoRA. See the table below on GSM8K. For instance, **our ReMix's accuracy 64.22 for $k=4$ significantly outperforms rank-32 LoRA's accuracy 59.21**. This clearly demonstrates that our method is able to choose different subsets appropriately.
>
> |Method|$k=1$|$k=2$|$k=4$|
> |:--|:-:|:-:|:-:|
> |Rank-$kr$ LoRA|56.10|54.51|59.21|
> |Ours: mixture of $k$ rank-$r$ LoRAs|56.18|59.67|64.22|
>
> > W3: The paper does not seem to discuss how the value of $k$ should be chosen.
>
> Thanks for the suggestion. The choice of $k$ depends on the tradeoff between efficiency and accuracy. Theoretically, since the number of size-$k$ subsets (i.e., $\binom nk$) increases with $k$ when $k\le n/2$, we would expect performance to increase with $k$ as well. Indeed, as shown in the following table (GSM8K), our ReMix consistently achieves stronger results under larger $k$ whenever $k\le n/2$.
>
> |$k=1$|$k=2$|$k=3$|$k=4$|
> |:-:|:-:|:-:|:-:|
> |56.18|59.67|61.33|64.22|
>
> > W4: The reported accuracy gains may correlate with the increased number of parameters rather than improved routing balance. The paper does not seem to disentangle these two effects or provide ablation results controlling for parameter count.
>
> Firstly, we'd like to clarify that **we have controlled the parameter** count for all methods in Table 1. We use the same parameter count budget for all methods, and for each method, we grid-search for the best parameter count and report the accuracy under the best parameter count. This ensures a fair comparison and clearly demonstrates that the improvement comes from our improved routing rather than the parameter count. For instance, on HumanEval, MixLoRA has pass@1=29.27 under 0.116B parameters while our ReMix achieves pass@1=32.93 under 0.090B parameters. That is, our method achieves **better performance under fewer parameters than MixLoRA**.

---

### Meta-Review · Area_Chair_WAhy · 2026-01-06

**Summary:**

After careful consideration, I recommend a reject due to: i) unclear motivation of the paper, ii) the lack of discussion over how some parameters should be chosen even after the authors' rebuttal, iii) lack of generalizability of the proposed solution, iv) it seems some of the author's claims during response is inconsistent; it is not obvious what was changed in the PDF. For example, where is the layerwise heatmap analysis?

**Reviewer Concerns:**

The reviewer's question on motivation is not yet resolved. Some reviewers also mentioned the problem with adjusting the value of k of which the authors does not respond well enough. The problem of routing imbalance is also not well resolved.

**Reviewer Scores:**

I do not think the reviewers will change the score a lot. At best, I predict something like a 6642.

---

### Decision · Program_Chairs · 2026-01-26

Reject